# Development of a Functional Cookie Formulated with Chaya (*Cnidoscolus aconitifolius (Mill*.) I.M. Johnst) and Amaranth (*Amaranthus cruentus*)

**DOI:** 10.3390/molecules27217397

**Published:** 2022-10-31

**Authors:** Azalia Avila-Nava, Sayuri L. Alarcón-Telésforo, José Moisés Talamantes-Gómez, Luis Corona, Ana Ligia Gutiérrez-Solis, Roberto Lugo, Claudia C. Márquez-Mota

**Affiliations:** 1Hospital Regional de Alta Especialidad de la Península de Yucatán, Mérida 97130, Mexico; 2Departamento de Nutrición Animal y Bioquímica, Facultad de Medicina Veterinaria y Zootecnia, Universidad Nacional Autónoma de México (FMVZ-UNAM), Ciudad de México 04510, Mexico

**Keywords:** functional cookie, fiber, vegetal protein, antioxidant activity, PUFAs

## Abstract

Chaya and amaranth are Mexican traditional foods with a high nutritional value. Many studies have demonstrated the individual beneficial effect of each. However, there is no evidence of the use of these foods on the formulation of functional foods. This study evaluated the effect of replacing 5–20% of wheat flour with chaya and amaranth flours to generate four different formulations of cookies. Proximal analysis, total polyphenols and oxalate content, antioxidant activity, fatty acid profile, and sensory analysis were performed on the cookies. The results of the chemical composition showed that all cookies have a high protein content (9.21–10.10%), an adequate amount of fiber (5.34–6.63%), and a balanced ratio of unsaturated–saturated fatty acids (70:20), and they contain PUFAs (50.4–53.2 g/100 g of fatty acids), especially α-linolenic and oleic acids. All formulations presented antioxidant activity (2540 ± 65.9 to 4867 ± 61.7 Trolox equivalents (μmoles/100 g)) and polyphenols (328–790 mg/100 g); in particular, quercetin was identified in their composition. Results of the sensory analysis indicated that incorporation of chaya and amaranth flour in cookies does not affect the acceptability of the products. The inclusion of traditional foods, such as chaya and amaranth, in cookies enhances their nutritional value and increases the content of bioactive compounds associated with health effects.

## 1. Introduction

Emerging data support traditional foods that connect ethnicity to dietary intake patterns to promote healthy lifestyles; meanwhile, there are pieces of evidence that showed a correlational relationship between acculturation and metabolic alterations, which are currently part of the main public health problems worldwide and nationally [1,2].

In this sense, nowadays, the goals of the United Nations Sustainable Development Group require food systems to be sustainable and equitable to eradicate malnutrition and hunger. It is important that food systems include different factors such as activities involved in food production, processing and storage, retail, trade, and consumption [3]. However, this goal is complicated due to the globalization that exists in different low- and medium-developed countries, in which acculturation has generated an increase in industrialized products, as well as a decrease in the consumption of traditional foods. Nowadays, there is an increase in the prevalence of metabolic diseases, such as obesity, metabolic syndrome, type 2 diabetes, and dyslipidemia, which are the major causes of death in the Mexican population [4].

Therefore, strategies have been sought to promote the consumption of traditional foods, thus encouraging the use of the region’s own resources. Inclusion of traditional foods in the diet has been associated with beneficial effects on health and as a strategy to control the development of metabolic diseases [5]. In Mexico, among these traditional foods are chaya and amaranth, which have been associated with various beneficial effects such as lowering the serum levels of glucose, triglycerides, cholesterol, and LDL-C, in addition to their antioxidant and anti-inflammatory effects [6,7]. One of the oldest and most abundant plants of the Yucatan Peninsula is chaya (*Cnidoscolus aconitifolius (Mill*.) I.M. Johnst). This plant has significant nutritional and medicinal properties in Mexico. Chaya is an edible leaf mainly grown and produced in Campeche, Yucatan, and Chiapas. It is an important component of the usual diet of indigenous communities due to its nutritional value because it contains dietary fibers, proteins, minerals, vitamins A and C, and polyphenols. Most abundant phenolic compounds identified in chaya are kaempferol and quercetin. Many studies have shown that chaya have important anti-inflammatory and antioxidant activities [8,9]. Another Mexican traditional food that has been associated with beneficial effects is amaranth (*Amaranthus cruentus*). This is one of the oldest known edible vegetables, found in Tehuacan, Puebla, Mexico. Amaranth is considered a “superfood” due to its high nutritional value. It is a gluten-free pseudo-cereal with high protein content (13–19%); it also is a source of fatty acids (FAs), especially polyunsaturated fatty acids (PUFAs), tocopherols, phenolic compounds, flavonoids, vitamins, and minerals [10]. Thus, amaranth has been associated with hepaprotective, antioxidant, antihyperglycemic, and hypolipidemic activities [11]. Despite the beneficial effects associated with these foods, to the best of our knowledge, there are few studies that use these to generate functional foods. Thus, the aim of the present study was to develop a functional cookie formulated with chaya and amaranth flours.

## 2. Results

### 2.1. Chemical Composition of Different Formulation of Cookies

To development a functional cookie, we replaced 5–20% of wheat flour with chaya and amaranth flours. The formulations generated were as follow: 40 g of wheat flour + 5 g of chaya flour + 20 g of amaranth flour (Ch5:A20); 40 g of wheat flour + 10 g of chaya flour + 15 g of amaranth flour (Ch10:A15); 40 g of wheat flour + 15 g of chaya flour + 10 g of amaranth flour (Ch15:A10); and 40 g of wheat flour + 20 g of chaya flour + 5 g of amaranth flour (Ch20:A5).The cookie serving size used in the present study was 30 g (Appendix A) with a similar appearance of physical characteristics using the different formulations of cookies (Figure 1A–D).

With regard to their chemical compositions, the four formulations have a protein content higher than 8% (Table 1). Formulation Ch10:A15 contains the highest content of protein and fiber. On the other hand, formulation Ch20:A5 showed the highest percentage of lipids compared with the other formulations (Table 1).

### 2.2. Antioxidant Activity and Phytochemical Profile of Different Formulation of Cookies

One characteristic of functional foods is the antioxidant activity caused by the presence of bioactive compounds, such as polyphenols. The results showed that all cookie formulations presented polyphenols (Figure 2A). It is observed that the highest concentration of polyphenols was in the Ch20:A5 formulation; its content is 58.5% more than that in formulation Ch10:A15, which is the sample with the lowest concentration of total polyphenols. These results could be associated with the presence of chaya in this formulation. With regard to the antioxidant activity, the results showed that the sample Ch20:A5 has the highest antioxidant activity (4867 ± 61.7 Trolox equivalents (μmoles/100 g)), and the lowest was presented in sample Ch5:A20 (2540 ± 65.9 Trolox equivalents (μmoles/100 g)) (Figure 2B).

The results about the phytochemical profile of the samples showed the presence of quercetin. The retention time of quercetin was determined at 6.39 min (Appendix A). The HPLC profile obtained showed a pattern with one peak that represented the quercetin present in different formulations of cookies. The formulation Ch20:A5 showed the highest concentration of quercetin with 2.61 ± 0.02 mg/100 g, followed by Ch5:A20 with 1.38 ± 0.02 mg/100 g, Ch15:A10 with 1.05 ± 0.02 mg/100 g, and Ch10:A15 with 0.84 ± 0.01 mg/100 g (Figure 3A–D).

### 2.3. Oxalate Content in Different Formulation of Cookies

An important evaluation is the oxalate concentration due to the fact that this compound is considered an antinutrient; thus, it is important to reduce its content in foods. The results showed that chaya flour contains the highest concentration of oxalates. However, this concentration decreased at least two times in the cookie formulations. The formulation contained the lowest oxalate content, which is half of formulation Ch15:A10 (Table 2).

### 2.4. Fatty Acid Profile of Different Formulation of Cookies

The cookies contain a high amount of unsaturated FAs compared with saturated FAs (Table 3). Formulation Ch20:A5 showed the best balance in these FAs with 73.19% of unsaturated FAs and 26.81% of saturated FAs. With regard to the content of PUFAs, the four formulations presented these types of FAs (Table 3). However, there is a difference in the percentage of chaya or amaranth. Formulations Ch10:A15 and Ch15:A10 have a higher content of linoleic and linolenic acids.

### 2.5. Microbial Analysis of Different Formulations of Cookies

The results of the microbial analysis showed that the presence of pathogenic microorganisms in the cookies is within the permissible limits according to the Mexican legislation [12] (Table 4).

### 2.6. Sensorial Analysis of Cookies

The result of the sensory analysis is presented in Table 5. Formulations Ch5:A20, Ch10:A15, and Ch15:A10 had the highest color score ranking (*p* > 0.05), and the formulations with the highest texture score ranking were Ch5:A20 and Ch15:A10 (*p* > 0.05); meanwhile, the formulation with the lowest texture ranking was Ch20:A5 (*p* > 0.05). According to the panelists, Ch10:A15, Ch15:A10, and Ch20:A5 showed the lowest sweetness score ranking (*p* > 0.05), and interestingly, there was no difference in the sweet aroma among the formulations. Ch20:A5 had the highest herbal score ranking among the formulations (*p* > 0.05). Finally, formulation Ch5:A20 had better taste in comparison with the other formulations (*p* > 0.05).

The overall score analysis is presented in Figure 4, which demonstrates that Ch5:A20 had a better score in comparison with Ch20:A5. Moreover, as previously described, Ch20:A5 had higher herbal aroma among the proposed formulations.

## 3. Discussion

Our results showed that the inclusion of chaya and amaranth flours for the development of a functional cookie promoted a high protein content, an adequate amount of fiber, and the PUFA content, especially α-linolenic and oleic acids. Additionally, they showed antioxidant activity and contained polyphenols, especially quercetin. The development of this type of products is important, since it promotes the use of traditional foods that can be utilized to generate easily accessible and low-cost health strategies to improve the health of the population.

The population in middle-income countries frequently present unhealthy eating patterns, which are related with the development of pathologies such as obesity, type 2 diabetes, dyslipidemia, hypertension, and cardiovascular alterations [13]. Thus, currently, the nutritional strategies to prevent these alterations could be the consumption of foods with fibers, vegetal proteins, antioxidant activity, and specific bioactive compounds, which can regulate or reduce the metabolic alterations.

The chemical composition of the cookies must be in accordance with some reference standards; one of them is the protein content. Our results showed that the cookies’ protein content was between 9% and 10%; this is in accordance with the National Nutrient Database for Standard Reference, which reported that the content of protein in butter cookies should be 6.10% [14]. Previous studies have demonstrated that the protein content of whole-wheat flour cookies is in the range of 6–12% [15,16,17]. Additionally, recent nutritional guidelines recommend to increase protein consumption and limit or decrease the consumption of refined grains, added sugars, and saturated fats [18]. It is important to mention that the protein in the cookie generated is mainly from chaya and amaranth. Vegetable protein is one of the compounds that generated controversy; for several years, these proteins were considered a low-quality protein because of the low content of their indispensable amino acids [19]. However, nowadays, there is an interest in incorporated plant-based proteins as dietary components to promote health effects such as regulation of lipid and glucose metabolism [20]. Incorporation of chaya, which contains leucine, lysine, and phenylalanine [21], and amaranth rich in lysine and sulfur containing amino acids [22] may be an alternative to supply the deficiency of these amino acids in wheat flour [23]. Thus, we can hypothesize that the inclusion of chaya and amaranth flours might improve the nutritional quality of butter cookies. However, further studies are recommended to assess the protein quality of the formulated cookies. Vegetal protein is related with beneficial effects because these types of foods also provide other nutrients to the diet such as fiber. Interestingly, the inclusion of chaya and amaranth flour to a traditional butter cookie recipe increased the fiber content by 6.6- to 7.8-folds in comparison with the fiber content reported in butter cookies [14]. These cookies are a good fiber source that might have health benefits to the consumers, since it has been well-demonstrated that the consumption of 25–35 g of fiber per day is associated with lower incidence of metabolic diseases and coronary heart diseases [24].

Among the molecular mechanisms involved in metabolic pathologies are oxidative stress (OS) and inflammation [25,26,27]. Thus, one of the nutritional strategies to prevent these alterations could be the consumption of foods with antioxidant activity and bioactive compounds. Our results showed that all formulations showed antioxidant activity, but the cookie with 20% of chaya flour presented with the highest activity. Similar results were previously reported in cookies made with flours from traditional foods; the results showed that cookies made with flour from nopal presented with the highest antioxidant activity (96.3 ± 0.01% of ABTS+ scavenging ability) followed by *P. ostreatus* flour (93.49 ± 2.61% of ABTS+ scavenging ability) and amaranth flour (77.52 ± 1.74% of ABTS+ scavenging ability). Other studies included flours from foods associated with antioxidant effects such as blueberry and spirulina. The use of blueberry flour showed that cookies presented with 187.46 ± 4.29 μM Trolox equivalents/g in dry weight basis [28]. On the other hand, the addition of 2% of spirulina flour to the formulation showed 1.20 ± 0.00 gallic acid equivalent per gram of sample [29].

These results suggested that the incorporation of foods with antioxidant compounds produces a beneficial effect on the composition of the developed product. However, the antioxidant effect depends on not only the concentration of compounds that confer this activity but also its bioavailability [30]. Previous reports have shown that the quantification of total polyphenols may be underestimated because most of the polyphenols are bound to dietary fiber and are not bioavailable to generate the antioxidant effect. In this sense, our results showed that the formulation with the highest amount of chaya flour is the one with the highest concentration of total polyphenols and quercetin. These characteristics may be mainly due to the presence of chaya. In fact, previous studies showed the antioxidant activity of chaya, as well as the presence of bioactive compounds such as quercetin in its composition [8,9,31].

Lipid profile is an important result of the composition from all cookie formulations. Our results showed that all formulations contained 17–18% of total lipids with a high amount of unsaturated FAs compared with saturated FAs. A similar amount of the total content of lipids in cookie formulations was previously reported [28,29,32]. However, it is important to mention that our study quantified and identified specific FAs, and previous studies did not specify the amount of saturated and unsaturated FAs. The quantification of these types of FAs has technological and health relevance. On the one hand, the presence of saturated FAs in the chemical composition of the cookies is related to the ease of handling during the manufacture and the desired crunchiness of the cookies [31]. In the area of health, high consumption of saturated FAs is not recommended due to their unhealthy effects. In fact, the Dietary Guidelines for Americans recommend limiting calories from saturated fats to less than 10% of the total calories you eat and drink each day [32].

As a result, the types of fats used in the manufacture of cookies have been replaced. One of these strategies is the increase in PUFAs due to the benefits they provide. PUFAs can modulate pathways related with lipid and carbohydrate metabolism. It has been reported that PUFAs promoted an increasing fatty acid oxidation through the activation of peroxisome proliferator-activated receptor alpha (PPAR-a) or reduced the activation of sterol-regulatory element binding protein-1c (SREBP-1c), which inhibited lipogenesis [33]. Additionally, the consumption of PUFAs can reduce CVD risk through the modulation of excessive inflammation and OS.

Our results showed that all formulations contain α-linolenic acid (C18:3, ω-3) and oleic acid (C18:1, ω-9). The formulations with the highest concentration of α-linolenic acid (52.4 ± 0.04 g/100 g of FA) and oleic acid (21.9 ± 0.139 g/100 g of FA) were Ch20:A5 and Ch15:A10, respectively. These results could be due to the chemical composition of amaranth containing unsaturated and saturated FAs, and the most abundant FA is linoleic acid, which comprises approximately 40% of all FAs, followed by oleic acid [34]. A study showed that the use of *Platonia insignis* nuts as an ingredient in the preparation of cookies reported 43.48 ± 0.37 % of oleic acid and 11.14 ± 0.15% of linolenic acid [35]. Another study also reported the presence of these FAs in cookies by the addition of flax seeds, which contained 4.75–5.31% of linolenic acid [36].

In the use of foods of plant origin, the identification of compounds considered as “antinutrients” must also be considered due to their adverse effects. One of these compounds is oxalate, which is present in chaya [37,38,39]. Excessive consumption of oxalates is contraindicated due to their ability to generate insoluble complexes with minerals such as calcium (Ca^2+^), iron (Fe^2+^), and magnesium (Mg^2+^) by promoting their accumulation [38]. In this sense, many strategies such as soaking, steaming, boiling, or cooking have been reported to reduce the oxalate concentration from foods. Our results showed a significant decrease in the oxalate content in the different formulations of cookies in comparison with the chaya flour, indicating that the baking temperature has an important role in decreasing this antinutritional component. Similar results reported a decrease of 30% and 87% in the total oxalate content of boiling vegetables compared with that of baked or steamed vegetables [39].

These thermal treatments are also related to the reduction of bacteria, which could generate harmful health effects. In this sense, the quantification of aerobic mesophilic microorganisms, total coliforms, yeasts, and molds is a measure of management conditions and good manufacturing practice; our results indicate that the management of the cookie ingredients was correctly performed.

As previously mentioned, chaya leaf is a plant with a high nutritional value, and it is widely traditionally consumed in beverages, soups, salads, and food preparations [40]; but there is no evidence of the use of chaya flour in the formulation of bakery products. On the other hand, amaranth is a pseudo-cereal that can be successfully integrated in bakery foods [41,42]. Thus, it is important to assess the acceptance of the mixture of chaya and amaranth flour in butter cookies. Food sensory analysis is fundamental in the development of new products; among these studies, hedonic testing is widely used to determine the acceptance or preference of a product by consumers [43]. In general, all the formulations had good sensorial score, but the most preferable one was Ch5:A20; this could be attributed to the higher content of amaranth flour. Meanwhile, we observed that cookies with Ch20:A5 had the lower acceptance values for color, texture, and taste and a higher value of herbal aroma among the formulations, indicating that the inclusion of 20 g chaya/100 g might cause lower acceptance of the cookies. The present study showed that the incorporation of flour from chaya and amaranth in cookie formulations did not promote a negative effect on the sensory and chemical properties of cookies. Thus, the product was able to have its own characteristics, in addition to achieving a good acceptance in its consumption.

In some countries, such as Mexico, ultraprocessed foods such as cookies are now one of the most consumed baking goods [44,45]. However, this type of products has been related to the high consumption of simple carbohydrates and saturated fats. In this sense, it is important to look for alternatives for the generation of similar products that provide a health benefit to the consumer and according to the new trends in the search for foods in the market that promote healthy eating. One of these strategies is to reintroduce the use of traditional, accessible, and low-cost food such as chaya and amaranth. These foods have great importance in the indigenous populations of different areas of the country: amaranth in central Mexico and chaya in the southeast. However, their use has been lost through generations due to changes in lifestyle. Thus, promoting the use of these foods for developing new products may not only impact health benefits by enhancing their individual characteristics but also contribute to generating new products with added value to the local markets that are part of the intangible heritage of the communities and are a fundamental part of the local and national economy. Additionally, development of these types of products had an impact on cultural, social, and technological areas.

However, it is important to note that more studies are needed to prove that the product really has the beneficial effects that it presents as chemical characteristics. The tests can be carried out with intervention studies in animal models and later transferred to a specific population, with the purpose of generating evidence about them and establishing nutritional strategies with the consumption of this type of products.

## 4. Materials and Methods

### 4.1. Samples

Chaya leaves (*Cnidoscolus aconitifolius* (Mill.) I.M. Johnst) was collected in Mérida, Yucatán, from March to June 2019. The qualified botanist José-L. Tapia-Muñoz identified the plant, and a voucher specimen (72343) was deposited at the herbarium of the Centro de Investigación Científica de Yucatán. To obtain chaya flour, the leaves were dried for 6 h at 57 °C (Novatech, Jalisco, México). After drying, the leaves were milled for 3 min and stored at room temperature until use. The amaranth flour and baking ingredients, i.e., whole-wheat flour, desalted butter, brown sugar, vanilla extract, and eggs, were purchased from a local market.

### 4.2. Formulation of Cookies with Chaya and Amaranth Flour

The cookie dough was elaborated according to a traditional butter cookie recipe with the modifications shown in Table 6. The levels of inclusion of chaya and amaranth flour were selected according to previous studies where it was reported that substitution of nonconventional flours with levels higher than 50% may cause changes in the flavor and consumer’s acceptance of the baking goods [8].

To make the cookie dough, the desalted butter was creamed and blended with sugar, egg, and vanilla extract; the mixture was set aside. The whole wheat, chaya, and amaranth flour were sieved and added to the previous mix; the batter was homogenized until a smooth dough was obtained; after mixing, the dough was refrigerated (4 °C for 30 min). After this time, the dough is cut into four different shapes and baked at 180 °C for 25 min; after baking, the cookies were cooled at room temperature for 30 min and stored at room temperature protected from the light in hermetic containers. Figure 2 shows representative photographs of the formulated cookies.

### 4.3. Analysis of Chemical Composition of Cookies

The chemical composition of the different samples was determined using the techniques recommended by the Association of Official Analytical Chemists (AOAC) [46].

### 4.4. Determination of Antioxidant Activity and Total Polyphenols

The antioxidant activity in different samples was determined by oxygen radical absorbance capacity (ORAC) [47]. The total polyphenol concentration was evaluated using the Folin–Ciocalteu method [48].

### 4.5. Evaluation of Phytochemical Profile of Cookies by HPLC

The phytochemical profile of the sample of cookies was analyzed with a high-performance liquid chromatography (HPLC) system (Agilent 7 Technologies 1100) using a C18 column (Thermo Scientific ODS Hypersil) (250 × 4.6 mm, 5.0 μm). For HPLC analysis, an extract from 2 g of the dry sample from each formulation was obtained. The sample extraction was adapted from previous reports [49]. Extraction was carried out using 2 g of the dry sample with 30 mL of ethanol (85%) at 60 kHz in an ultrasonic extraction device for 15 min and repeated three times. The extract was stored at −4 °C overnight. Then, 15 mL of methanol and 1.5 mL of hydrochloric acid were added and left to reflux for 2 h at 60 °C. The solvent was removed on a rotary evaporator. The dried extract was dissolved in methanol; approximately 1 mL of the extraction was filtered through a 0.45 μm membrane and transferred into HPLC vials. The sample was eluted at a 1 mL/min flow rate with CH_3_OH: CH_3_CN: H_2_O (40:15:45) as a gradient mobile phase with 1% of acetic acid and detected by absorbance at 368 nm using a Waters 2998 photodiode array detector (13). The calibration curves of quercetin and kaempferol (Sigma, St. Louis, MO, USA) were constructed using serial dilutions in ethanol of the standard solution (0.01, 0.05, 0.15, and 0.30 mg/mL). The sample was analyzed in sextuplicate.

### 4.6. Determination of Oxalates Content

The content of oxalates in the samples was evaluated through the extraction of oxalic acid followed by its oxidation to CO_2_, which was quantified by titration [50].

### 4.7. Quantification of Fatty Acid Profile

For the determination of fatty acids, the oil was extracted from the samples; the fatty acids were derivatized and, finally, analyzed by gas chromatography coupled to mass spectrometry (GC/MS). Total lipids from the samples were extracted by the Soxhlet method and methylated as previously described [51]. The methylated fatty acids were analyzed by gas chromatography (Perkin Elmer AutoSystem XL with FID detector, Waltham, Massachussetts, United States) with a flame ionization detector (Agilent) using a column (select FAME CP-7420 100 m × 0.25 mm × 0.25 um) (Agilent Technologies, Santa Clara, CA, USA).

### 4.8. Microbiological Analysis of Cookies

The microbiological analysis of the cookies was carried out in agreement with the Mexican Official Standards NOM-247-SSA1-2008 [12] in the Laboratorio de microbiología e inmunología del Departamento microbiología e inmunología of the Facultad de Medicina Veterinaria y Zootecnia.

### 4.9. Sensorial Analysis of Cookies

Analysis of samples of cookies by panel constituted of a group of voluntary tasters (n = 50). The sensory analysis was performed using a 9-point hedonic scale for color, taste, and sweetness ((1)—“disliked extremely”, (2)—“disliked very much”, (3)—“disliked moderately”, (4)—“disliked slightly”, (5)—“neither liked nor disliked”, (6)—“liked slightly”, (7)—“liked moderately”, (8)—“liked very much”, and (9)—“liked extremely”); and a 5-point hedonic scale for sweet aroma, herbal aroma ((1)—“imperceptible”, (2)—“perceptible”, (3)—“slightly distinctive”, (4)—“distinctive”, (5)—“ distinctive”)), and texture ((1)—“very poor”, (2)—“poor”, (3)—“fair”, (4)—“good”, (5)—“very good”)) [52].

The sensory test was performed in a room under natural white light at room temperature. The different samples of cookies were coded and presented on a white plate. The volunteers were instructed to consume the samples from right to left; moreover, water was provided for rinsing the mouth between samples. Each sample was individually evaluated by descriptive grades using the scales provided.

### 4.10. Statistical Analysis

The Smirnov test was performed to analyze the type of distribution. Differences between samples were evaluated by one-way ANOVA followed by the Tukey test. A value of *p* < 0.05 was considered significant. Data were presented as the mean ± standard error of the mean (SEM). Data were analyzed using RStudio 2021.09.0.

## 5. Conclusions

The inclusion of traditional foods, such chaya and amaranth, in cookies enhances the nutritional value and increases the content of bioactive compounds associated with health effects.

## Figures and Tables

**Figure 1 molecules-27-07397-f001:**
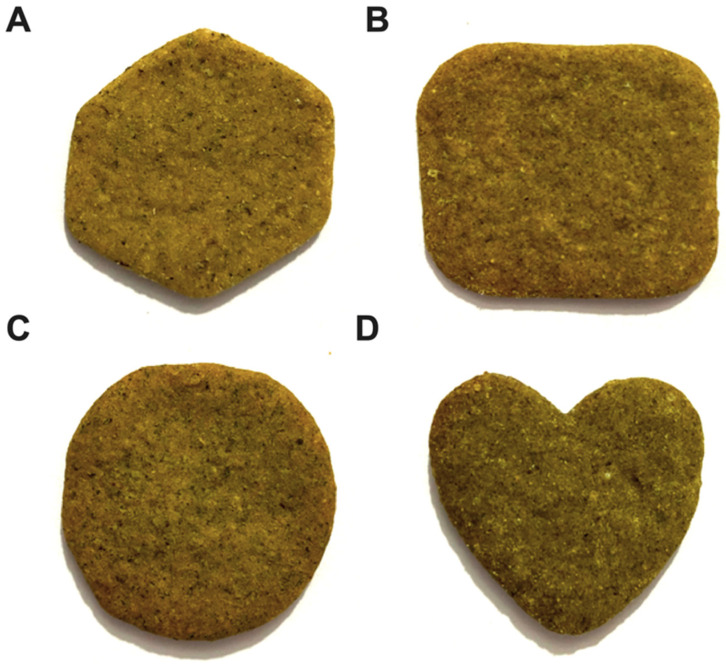
Representative photographs of different formulations of cookies. (**A**) Ch5:A20: 40 g of wheat flour + 5 g of chaya flour + 20 g of amaranth flour. (**B**) Ch10:A15: 40 g of wheat flour + 10 g of chaya flour + 15 g of amaranth flour. (**C**) Ch15:A10: 40 g of wheat flour + 15 g of chaya flour + 10 g of amaranth flour. (**D**) Ch20:A5: 40 g of wheat flour + 20 g of chaya flour + 5 g of amaranth flour.

**Figure 2 molecules-27-07397-f002:**
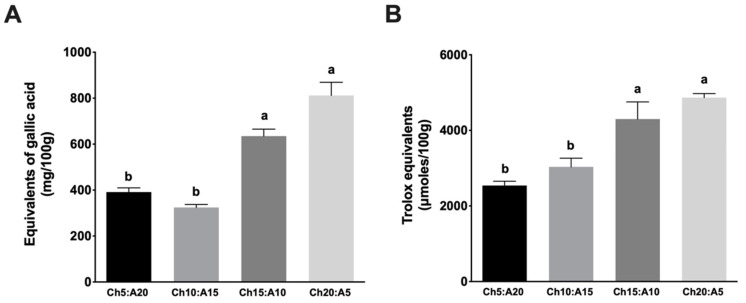
Total polyphenols and antioxidant activity of different formulations of cookies. (**A**) Concentration of total polyphenols. (**B**) Antioxidant activity. Values are mean ± SD. Statistical differences are indicated by different letters (a > b). Value of *p* < 0.05 is considered significant. Ch5:A20: 40 g of wheat flour + 5 g of chaya flour + 20 g of amaranth flour; Ch10:A15: 40 g of wheat flour + 10 g of chaya flour + 15 g of amaranth flour; Ch15:A10: 40 g of wheat flour + 15 g of chaya flour + 10 g of amaranth flour; and Ch20:A5: 40 g of wheat flour + 20 g of chaya flour + 5 g of amaranth flour.

**Figure 3 molecules-27-07397-f003:**
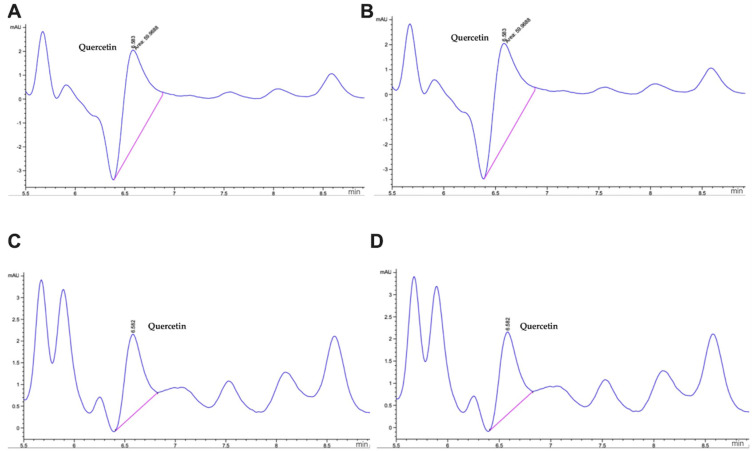
Phytochemical profile by HPLC. Principal compound in different formulations of cookies is quercetin (6.39 min). (**A**) Ch5:A20: 40 g of wheat flour + 5 g of chaya flour + 20 g of amaranth flour; (**B**) Ch10:A15: 40 g of wheat flour + 10 g of chaya flour + 15 g of amaranth flour; (**C**) Ch15:A10: 40 g of wheat flour + 15 g of chaya flour + 10 g of amaranth flour; and (**D**) Ch20:A5: 40 g of wheat flour + 20 g of chaya flour + 5 g of amaranth flour.

**Figure 4 molecules-27-07397-f004:**
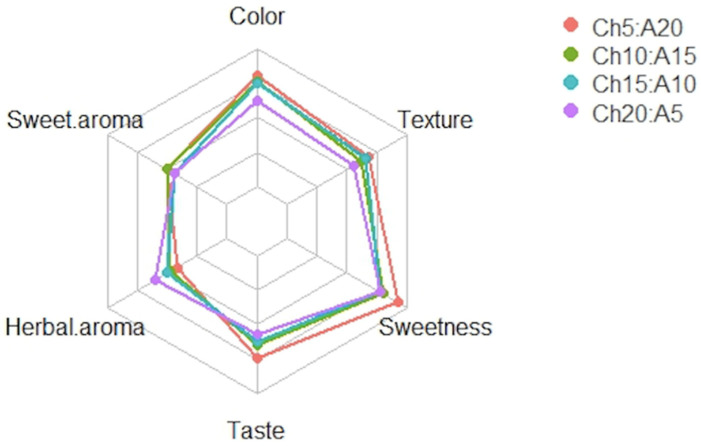
Overall sensorial analysis of different formulations of cookies. Ch5:A20: 40 g of wheat flour + 5 g of chaya flour + 20 g of amaranth flour; Ch10:A15: 40 g of wheat flour + 10 g of chaya flour + 15 g of amaranth flour; Ch15:A10: 40 g of wheat flour + 15 g of chaya flour + 10 g of amaranth flour; and Ch20:A5: 40 g of wheat flour + 20 g of chaya flour + 5 g of amaranth flour.

**Table 1 molecules-27-07397-t001:** Chemical composition of different formulations of cookies. Data are presented as g in 100 g.

Component	Formulation			
	Ch5:A20	Ch10:A15	Ch15:A10	Ch20:A5
Moisture	2.86 ± 0.03 ^c^	3.77 ± 0.07 ^b^	4.36 ± 0.10 ^a^	2.90 ± 0.003 ^c^
Ash	0.03 ± 0.02 ^c^	0.91 ± 0.01 ^c^	1.03 ± 0.02 ^b^	1.13 ± 0.00 ^a^
Protein	10.1 ± 0.00 ^b^	10.6 ± 0.02 ^a^	10.5 ± 0.16 ^a^	9.21 ± 0.01 ^c^
Lipids	18.0 ± 0.05 ^ab^	17.4 ± 0.32 ^b^	17.7 ± 0.03 ^ab^	18.2 ± 0.04 ^a^
Fiber	6.32 ± 0.07 *	6.63 ± 0.24 *	5.34 ± 0.57 *	5.63 ± 0.10 *
Sodium	7.53 ± 0.17 ^a^	4.99 ± 0.12 ^a^	3.46 ± 0.07 ^c^	4.07 ± 0.02 ^b^

Values are means ± SD. Statistical differences are indicated by different letters (a > b > c). Value of *p* < 0.05 is considered significant. * These data were done in duplicate, so their statistical analysis was not performed. Ch5:A20: 40 g of wheat flour + 5 g of chaya flour + 20 g of amaranth flour; Ch10:A15: 40 g of wheat flour + 10 g of chaya flour + 15 g of amaranth flour; Ch15:A10: 40 g of wheat flour + 15 g of chaya flour + 10 g of amaranth flour; and Ch20:A5: 40 g of wheat flour + 20 g of chaya flour + 5 g of amaranth flour.

**Table 2 molecules-27-07397-t002:** Oxalate content in different formulations of cookies. Data are expressed as g of oxalic acid/g of sample.

Flour	Oxalate Content
Chaya	2.10 ± 0.27
Ch5:A20	0.31 ± 0.003 ^b^
Ch10:A15	0.70 ± 0.13 ^a^
Ch15:A10	0.39 ± 0.06 ^ab^
Ch20:A5	0.64 ± 0.02 ^a^

Values are mean ± SD. Statistical differences are indicated by different letters (a > b). Value of *p* < 0.05 is considered significant. Ch5:A20: 40 g of wheat flour + 5 g of chaya flour + 20 g of amaranth flour; Ch10:A15: 40 g of wheat flour + 10 g of chaya flour + 15 g of amaranth flour; Ch15:A10: 40 g of wheat flour + 15 g of chaya flour + 10 g of amaranth flour; and Ch20:A5: 40 g of wheat flour + 20 g of chaya flour + 5 g of amaranth flour.

**Table 3 molecules-27-07397-t003:** Fatty acid profile of different formulations of cookies. Data are presented as g in 100 g of FA.

Component	Formulation			
	Ch5:A20	Ch10:A15	Ch15:A10	Ch20:A5
Caproic acid (C6:0)	0.06 ± 0.001	0.08 ± 0.0017	0.1 ± 0.003	0.08 ± 0.004
Caprylic acid (C8:0)	0.26 ± 0.003	0.26 ± 0.0003	0.2 ± 0.001	0.23 ± 0.011
Capric acid (C10:0)	0.23 ± 0.0021	0.23 ± 0.0009	0.23 ± 0.003	0.19 ± 0.007
Lauric acid (C12:0)	3.85 ± 0.152	3.66 ± 0.003	3.68 ± 0.005	3.64 ± 0.134
Myristic acid (C14:0)	1.14 ± 0.046	1.04 ± 0.003	1.09 ± 0.027	1.0 ± 0.022
Palmitic acid (C16:0	13.97 ± 0.146	13.3 ± 0.070	13.4 ± 0.025	13.28 ±0.022
Palmitoleic acid (C16:1)	0.23 ± 0.002	0.21 ± 0.001	0.2 ± 0.002	0.24 ± 0.0003
Heptadecanoic acid (C17:0)	0.06 ± 0.001	0.06 ± 0.001	0.07 ± 0.001	0.07 ± 0.0009
cis-10Heptadecenoic acid (C17:1)	0.03 ± 0.0006	0.07 ± 0.0002	0.05 ± 0.00	ND
Stearic acid (C18:0)	5.9 ± 0.028	5.5 ± 0.013	6.08 ± 0.020	5.45 ± 0.018
Oleic acid (C18:1, ω-9)	21.1 ± 0.091	21.3 ± 0.022	18.6 ± 0.014	21.90 ±0.139
Linoleic acid (C18:2, ω-6)	0.65 ± 0.006	0.76 ± 0.007	0.74 ± 0.006	0.69 ± 0.026
α-Linolenic acid (C18:3, ω-3)	49.5 ± 0.237	50.5 ± 0.0004	52.4 ± 0.04	50.15 ± 0.350
γ-Linolenic acid (C18:3, ω-3)	0.07 ± 0.001	0.10 ± 0.003	0.08 ± 0.002	ND
Arachidic acid (C20:0)	2.2 ± 0.002	2.23 ± 0.009	2.72 ± 0.012	2.40 ± 0.007
Eicosanoid acid (C20:1, ω-9)	0.11 ± 0.002	ND	ND	0.11 ± 0.000
Heneicosylic acid (C21:0)	ND	0.07 ± 0.0004	0.05 ± 0.001	0.07 ± 0.003
Arachidonic acid (C20:4, ω-6)	0.14 ± 0.0013	0.09 ± 0.002	ND	ND
Behenic (C22:0)	0.27 ± 0.005	0.27 ± 0.002	0.20 ± 0.000	0.27 ± 0.0014
Eicosapentaenoic acid (C20:5, ω-3)	0.10 ± 0.002	0.11 ± 0.001	ND	0.10 ± 0.0003
Lignoceric acid (C24:0)	0.13 ± 0.001	0.13 ± 0.002	0.1 ± 0.0002	0.13 ± 0.0018
Σ SFA	28.08 ± 1.28	26.8 ± 1.12	27.9 ± 1.15	26.81 ± 1.12
Σ UFA	71.92 ± 8.16	73.2 ± 8.35	72.1 ± 8.62	73.19 ± 8.37
Σ MFA	21.45 ± 5.24	21.6 ± 7.06	18.8 ± 6.15	22.25 ± 7.24
Σ PUFA	50.46 ± 9.85	51.6 ± 10.1	53.2 ±17.3	50.94 ± 16.59

Values are mean ± SD. Ch5:A20: 40 g of wheat flour + 5 g of chaya flour + 20 g of amaranth flour; Ch10:A15: 40 g of wheat flour + 10 g of chaya flour + 15 g of amaranth flour; Ch15:A10: 40 g of wheat flour + 15 g of chaya flour + 10 g of amaranth flour; and Ch20:A5: 40 g of wheat flour + 20 g of chaya flour + 5 g of amaranth flour.

**Table 4 molecules-27-07397-t004:** Microbial count of mesophilic microorganisms, total coliforms, yeast, and molds of different formulation of cookies.

Microorganism	Reference Value * (CFU/g)	Formulation			
		Ch5:A20	Ch10:A15	Ch15:A10	Ch20:A5
Aerobic mesophilic microorganisms	3000	5	20	1	7.5
Total coliforms	<10	<3	<3	<3	<3
Yeast	300	<100	<100	<100	<100
Molds	300	<100	<100	<100	<100

Values are in CFU/g. CFU: colony-forming units; Ch5:A20: 40 g of wheat flour + 5 g of chaya flour + 20 g of amaranth flour; Ch10:A15: 40 g of wheat flour + 10 g of chaya flour + 15 g of amaranth flour; Ch15:A10: 40 g of wheat flour + 15 g of chaya flour + 10 g of amaranth flour; and Ch20:A5: 40 g of wheat flour + 20 g of chaya flour + 5 g of amaranth flour. * Reference value from the Mexican legislation N.O.M. 247-SSA1-2008 [12].

**Table 5 molecules-27-07397-t005:** Sensory evaluation of different formulations of cookies.

Sensory Attribute	Formulation			
	Ch5:A20	Ch10:A15	Ch15:A10	Ch20:A5
Color	6.4 ± 0.27 ^a^	6.1 ± 0.21 ^a^	6.0 ± 0.23 ^a^	5.0 ± 0.25 ^b^
Texture	3.4 ± 0.10 ^a^	3.1 ± 0.10 ^ab^	3.3 ± 0.08 ^a^	2.8 ± 0.12 ^b^
Sweetness	7.4 ± 0.19 ^a^	6.4 ± 0.24 ^b^	6.2 ± 0.22 ^b^	6.2 ± 0.23 ^b^
Sweet aroma	2.5 ± 0.10	2.5 ± 0.12	2.2 ± 0.12	2.2 ± 0.16
Herbal aroma	2.1 ± 0.12 ^b^	2.4 ± 0.15 ^b^	2.5 ± 0.15 ^ab^	3.0 ± 0.19 ^a^
Taste	5.9 ± 0.38 ^a^	5.2 ± 0.33 ^ab^	5.0 ± 0.28 ^ab^	4.6 ± 0.30 ^b^

Panelist *n* = 50; values are mean ± SD. Statistical differences are indicated by different letters (a > b). Value of *p* < 0.05 is considered significant. Ch5:A20: 40 g of wheat flour + 5 g of chaya flour + 20 g of amaranth flour; Ch10:A15: 40 g of wheat flour + 10 g of chaya flour + 15 g of amaranth flour; Ch15:A10: 40 g of wheat flour + 15 g of chaya flour + 10 g of amaranth flour; and Ch20:A5: 40 g of wheat flour + 20 g of chaya flour + 5 g of amaranth flour.

**Table 6 molecules-27-07397-t006:** Formulation of functional cookies with different concentrations of chaya and amaranth flours (g/100 g).

Ingredient (g/100 g)	Formulation			
	Ch5:A20	Ch10:A15	Ch15:A10	Ch20:A5
Whole-wheat flour	40			
Chaya flour	5	10	15	20
Amaranth flour	20	15	10	5
Brown sugar	17.5			
Desalted butter	17.5			
Egg *	1 piece (60)			
Vanilla extract	1			

* Vanilla extract and weight of the egg were not included in the total weight of the formulations.

## Data Availability

The data presented in this study are available upon reasonable request. Requests for data should be addressed to corresponding author.

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
