# Peer review of "Development of a Functional Cookie Formulated with Chaya (*Cnidoscolus aconitifolius (Mill*.) I.M. Johnst) and Amaranth (*Amaranthus cruentus*)"

_molecules, 2022, doi:10.3390/molecules27217397_

Round 1
Reviewer 1 Report
Overall, the manuscript is well planned study, however, following suggestions should be carefully considered.
Introduction: Please mention the chemical composition of the traditional foods from literature and include the novelty aspect at the end of manuscript. Why are these traditional foods required to include in cookies? If people have their availability in the region.
From the figure 1 cookies do not look of appropriate texture, authors should incorporate the texture analysis of cookies in this study.
Why the Ch20:A5 formulation had almost 900 mg GAE/g of cookie?? Which means every component was responsible for TPC???
Table 5, As mentioned above texture scores are very poor for the all formulations, which means the recipe could have been improved by adding some additive or additional component???
In sensory analysis an overall score category could have also been incorporated.
Author Response
Reviewer 1
Overall, the manuscript is well planned study, however, following suggestions should be carefully considered.
- Introduction: Please mention the chemical composition of the traditional foods from literature and include the novelty aspect at the end of manuscript. Why are these traditional foods required to include in cookies? If people have their availability in the region.
R1. As you suggested we added information about chemical composition in introduction section (lines 57-63 and 66-70). Also, we included a paragraph at the end of manuscript about the novel product and the health importance to generate this product from traditional Mexican foods (lines 370-384).
- From the figure 1 cookies do not look of appropriate texture, authors should incorporate the texture analysis of cookies in this study.
R2. Thank you for the suggestion, however in the present study it was not possible to perform the texture analysis of the cookies. Thus, the data obtained about this parameter was assessed through the sensory analysis. We will include in further studies the suggested analysis.
- Why the Ch20:A5 formulation had almost 900 mg GAE/g of cookie?? Which means every component was responsible for TPC???
R3. We apologize, this was a mistake. Data is mg GAE/100g, we corrected the data in the figure 2 and in the paragraph (lines 122-123). Content of TPC may be responsible mainly of Chaya flour; this result is according to our previous report that showed that an aquose extract with 40g of Chaya contained 1359 ± 47.2 mg GAE/L (Guevara-Cruz et al., 2021)
- Table 5, As mentioned above texture scores are very poor for the all formulations, which means the recipe could have been improved by adding some additive or additional component???
R4. For the texture score we use a five point hedonic scale ((1)—“very poor”,(2)—“poor”, (3)—“fair”, (4)—“good”,(5)—“very good”)) an the average respond was fair, there are several strategies to improve the cookies texture. As you kindly suggest, to improve the cookie texture, we can use different additives such as fat and sugar replacement such as polydextrose, a water-soluble additive that provide appropriate texture properties in baking goods (Mieszkowska & Marzec, 2016).
- In sensory analysis an overall score category could have also been incorporated.
R5. As you suggested we added a paragraph and figure of overall score of sensory analysis (Figure 4, lines 223- 225).
References
Elizabeth, L., Machado, P., Zinöcker, M., Baker, P., & Lawrence, M. (2020). Ultra-Processed Foods and Health Outcomes: A Narrative Review. Nutrients, 12(7), 1955. https://doi.org/10.3390/nu12071955
Marrón-Ponce, J. A., Flores, M., Cediel, G., Monteiro, C. A., & Batis, C. (2019). Associations between Consumption of Ultra-Processed Foods and Intake of Nutrients Related to Chronic Non-Communicable Diseases in Mexico. Journal of the Academy of Nutrition and Dietetics, 119(11), 1852–1865. https://doi.org/10.1016/j.jand.2019.04.020
Guevara-Cruz, M., Medina-Vera, I., Cu-Cañetas, T. E., Cordero-Chan, Y., Torres, N., Tovar, A. R., Márquez-Mota, C., Talamantes-Gómez, J. M., Pérez-Monter, C., Lugo, R., Gutiérrez-Solis, A. L., & Avila-Nava, A. (2021). Chaya Leaf Decreased Triglycerides and Improved Oxidative Stress in Subjects With Dyslipidemia. Frontiers in Nutrition, 8.https://www.frontiersin.org/article/10.3389/fnut.2021.666243
Mieszkowska, A., & Marzec, A. (2016). Effect of polydextrose and inulin on texture and consumer preference of short-dough biscuits with chickpea flour. Lwt, 73, 60–66.

Reviewer 2 Report
This is an interesting manuscript but can be improved.
Give the serving size in the abstract. I am assuming the serving size is 100 g which is very large for a cookie. In the US, the Food and Drug Administration (FDA) defines the serving size weight of a cookie as 30 g.
A Nutrition Facts label should be included in the manuscript to be able to compare the nutrients to commercially available cookies.
Line 186, what pathogens were assayed for? It looks like just standard plate counts were done.
Line 221, the authors compare their cookies to butter cookies, was this comparison done on a gram basis?
Table 6, the ingredients add up to greater than 100 g. The weight of the egg should be given in grams.
Line 381, what do the authors mean by “from beverage”
Line 447, there should have been an institutional review board review since the manuscript used human subjects in the sensory analysis.
Author Response
Reviewer 2
This is an interesting manuscript but can be improved.
1.Give the serving size in the abstract. I am assuming the serving size is 100 g which is very large for a cookie. In the US, the Food and Drug Administration (FDA) defines the serving size weight of a cookie as 30 g.
R1. As you suggested we added the serving size in grams of samples in results section (line 82) and in the supplementary table 1.
- A Nutrition Facts label should be included in the manuscript to be able to compare the nutrients to commercially available cookies.
R2. As you suggested we included the nutrition fact label of the product as supplemental table 1.
- Line 186, what pathogens were assayed for? It looks like just standard plate counts were done.
R3. The specifications of the Mexican legislation on products such as cookies include the microbiological analysis of Aerobic mesophilic microorganisms (<10 000 CFU/g) used the peptone yeast glucose agar, Total coliforms (< 10 CFU/g) used the violet red bile (VRBA) agar, Yeast (300 CFU/g) used the Sabouraud Dextrose agar and Molds (300 CFU/g) used the dichloran rose bengal agar (NOM, 2008, p. 247).
- Line 221, the authors compare their cookies to butter cookies, was this comparison done on a gram basis?
R4. To compare the nutritional value of the cookies we used as a reference the USDA National Nutrient Database for Standard Reference, making the comparison on 100g basis, this is due that the actual Mexican legislation state that food facts should be reported per 100 of product (Reynoso et al., 2021).
- Table 6, the ingredients add up to greater than 100 g. The weight of the egg should be given in grams.
R5. We did not consider vanilla extract and weight of egg in total amount of formulations. As you suggested we added the weight of the egg to grams in table 6. We did not include the weight of the egg and considered it as a piece because the addition of egg in cookies, biscuits and pastry is considered as an optional ingredient and its omission does not affect the product quality (Hui et al., 2008).
- Line 381, what do the authors mean by “from beverage”
R6. Thank you for your comment, it was a mistake, we have already modified the information (line 434).
- Line 447, there should have been an institutional review board review since the manuscript used human subjects in the sensory analysis.
R7. Respect to your questions, we only obtained verbal consent in sensorial analysis. According to Council for International Organizations of Medical Sciences (CIOMS) International Ethics Guidelines state that 'when the research design involves no more than minimal risk informed consent may be applied when a study only involves recording data from the medical record or administering a questionnaire that poses low risk (Eccles et al., 2011).
References
Eccles, M. P., Weijer, C., & Mittman, B. (2011). Requirements for ethics committee review for studies submitted to Implementation Science. In Implementation science (Vol. 6, Issue 1, pp. 1–3). BioMed Central.
Hui, Y. H., Corke, H., De Leyn, I., Nip, W.-K., & Cross, N. A. (2008). Bakery products: Science and technology. John Wiley & Sons.
Reynoso, I. J. C., Estrada, R. O., Paredes, L. V., Sancho, L. E. G. G., & Villaverde, I. de J. T. (2021). Actualización de la Norma Oficial Mexicana NOM-051-SCFI/SSA1 en la Industria de Alimentos y Bebidas. Journal Boliviano de Ciencias, 17(Especial), 6–18.
NOM, N. O. M. (2008). 247-SSA1-2008, Productos y servicios. Cereales y Sus Productos. Cereales, Harinas de Cereales, Sémolas o Semolinas. Alimentos a Base de: Cereales, Semillas Comestibles, de Harinas, Sémolas o Semolinas o Sus Mezclas. Productos de Panificación. Disposiciones y Especificaciones Sanitarias y Nutrimentales. Métodos de Prueba.

Round 2
Reviewer 1 Report
The manuscript has been revised as advised.